# Eukaryotic association module in phage WO genomes from *Wolbachia*

Sarah R. Bordenstein[1] & Seth R. Bordenstein[1,2]

Viruses are trifurcated into eukaryotic, archaeal and bacterial categories. This domain-specific ecology underscores why eukaryotic viruses typically co-opt eukaryotic genes and bacteriophages commonly harbour bacterial genes. However, the presence of bacteriophages in obligate intracellular bacteria of eukaryotes may promote DNA transfers between eukaryotes and bacteriophages. Here we report a metagenomic analysis of purified bacteriophage WO particles of *Wolbachia* and uncover a eukaryotic association module in the complete WO genome. It harbours predicted domains, such as the black widow latrotoxin C-terminal domain, that are uninterrupted in bacteriophage genomes, enriched with eukaryotic protease cleavage sites and combined with additional domains to forge one of the largest bacteriophage genes to date (14,256 bp). To the best of our knowledge, these eukaryotic-like domains have never before been reported in packaged bacteriophages and their phylogeny, distribution and sequence diversity imply lateral transfers between bacteriophage/prophage and animal genomes. Finally, the WO genome sequences and identification of attachment sites will potentially advance genetic manipulation of *Wolbachia*.

[1] Department of Biological Sciences, Vanderbilt University, Nashville, Tennessee 37232, USA. [2] Department of Pathology, Microbiology, and Immunology, Vanderbilt University, Nashville, Tennessee 37232, USA. Correspondence and requests for materials should be addressed to Seth R.B. (email: s.bordenstein@vanderbilt.edu) or to Sarah R.B. (email: sarah.bordenstein@vanderbilt.edu).

Viruses are the most abundant and diverse biological entities in the biosphere[1,2]. Infecting organisms across the tree of life, they associate with every ecosystem on the planet. They are generally classified into polythetic groups according to ecological niche and mode of replication[3,4]. While any cellular domain can be infected by a virus, no extant virus is known to traverse more than one domain[5,6]. This domain-specific ecology of viruses underpins the current taxonomic paradigm of trifurcating viruses into eukaryotic, archaeal and bacterial categories, along with recent reappraisals of whether viruses constitute a fourth domain of life[7,8]. As a result of this domain-specific ecology, viruses often integrate host genes via specific highways of lateral gene transfer. Eukaryotic viruses tend to hijack genes directly from their eukaryotic hosts to evade, manipulate and counter-strike antiviral immune responses[9,10], with the exception of some giant viruses that appear to acquire genes from all domains of life[11]. Bacterial viruses, or bacteriophages (phages), integrate genetic material from their bacterial hosts, including toxin[12], photosynthesis[13] and pigment biosynthesis genes[14] that contribute to the fitness of their bacterial host. To date, however, there is no archetypal case of phage particles harbouring genomes with eukaryotic DNA.

While all viruses are specific to one of the three domains of life, some bacteriophages target obligate intracellular bacteria of eukaryotic cells. For instance, phage WO infects the obligate intracellular alpha-proteobacteria *Wolbachia*, which in turn infect an estimated 40% of the most speciose group of animals worldwide—arthropods (as well as filarial nematodes). *Wolbachia* cause a range of host reproductive pathologies[15,16], primarily infect the cells of host reproductive tissues, exist in Golgi-derived vesicles within the eukaryotic cytoplasm, and are enclosed by a bacterial cell membrane and one or more eukaryotic-derived membranes[17,18]. Nearly all sequenced *Wolbachia* genomes, with the exception of those acting as obligate mutualists, harbour

prophage WO[19–21]. The prophage WO encode conserved structural modules (for example, head, tail and baseplate) and exhibit Caudovirales morphology in electron micrographs of purified phages[20,22–25]. Electron microscopy and quantitative analyses indicate that prophages undergo a lytic phase capable of rupturing bacterial and eukaryotic cell membranes, and phage WO occurs in the extracellular matrix of arthropod gonads[23,26]. Therefore, phage WO appears to uniquely contend with the cellular exit, entry and defence mechanisms of two separate domains of life. WO is also a promising tool for genome editing of *Wolbachia* that has thus far been refractory to genetic modification.

Here we assemble the sequenced genomes of phage WO particles, resolve the bacteriophage attachment and bacterial integration sites, report a eukaryotic association module in bacteriophages and discuss lateral gene transfers between eukaryotes and bacteriophages.

## Results

**Phage WO genomes reveal a eukaryotic association module.** Here we report the metagenomic analysis of phage WO particles from *w*VitA-infected *Nasonia giraulti* wasps and *w*CauB-infected *Ephestia kuehniella* moths (the *w*-prefix indicates specific *Wolbachia* strain and WO prefix indicates phage haplotype; see Supplementary Table 1 for a complete list). We identify the phage attachment sites and insertion regions and show from fully sequenced genomes that WO harbours all formerly described phage genetic modules (lysogeny, baseplate, head, replication, virulence, tail and patatin-like phospholipase[27]), as well as a new group of genes with atypical protein domains indicative of eukaryotic interaction. We collectively group these genes, which include the second largest gene in bacteriophages to date, into a 'eukaryotic association module' (EAM; Fig. 1, white box). The EAM

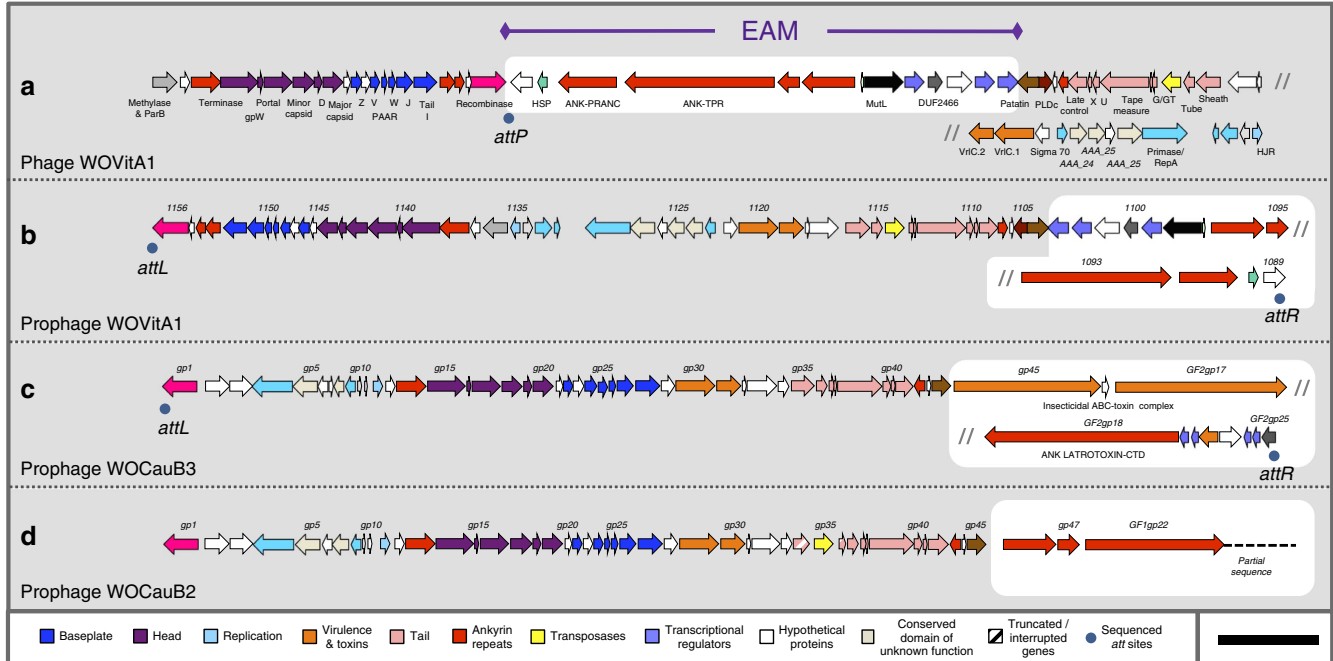

**Figure 1 | Phage WO genomes harbour an EAM.** The complete phage WO genome for (**a**) WOVitA1 was sequenced directly from purified viral particles using high throughput, metagenomic sequencing. The prophage (**b**) WOVitA1, (**c**) WOCauB3 and (**d**) WOCauB2 genomes were reannotated based on sequencing reads obtained from purified particles; complete genomes of WOCauB3 and WOCauB2 were not obtained. Each genome consists of a bacteriophage-like region (recombinase to patatin) and EAM highlighted in white. Grey slash marks indicate illustrative continuation of the genome. Dark blue dots indicate the discovery of the *attL* and *attR* sites of the prophage, which adjoin in the packaged WO genome to form *attP*. Numbers above the open reading frames indicate locus tags. Scale bar, 5,000 base pairs.

features genes that (i) encode homologous protein domains and cleavage sites central to eukaryotic functions, (ii) occur in phage and metazoan hosts, (iii) are among the largest genes in phage genomes (up to 14,256 bp) and (iv) are absent from mutualistic, phage-free genomes such as the bedbug-infecting wCle and filarial nematode-infecting wBm and wOo. They occur in all complete prophage WO haplotypes (Supplementary Table 2).

To verify the newly discovered EAM in the phage genome, we identified the terminal prophage WO genes and Sanger-sequenced amplicons from an independent sample of phage WOVitA1 (Fig. 1a) across the linear phage attP site (hypothetical protein gwv_1089 to recombinase, Supplementary Fig. 1). Next, using the newly identified attR and attL sites, we extrapolated the bacterial attB site in WOVitA1, which is a noncoding, repetitive sequence in Wolbachia from Nasonia wasps (Supplementary Fig. 1e). The full length of the completely assembled, linear WOVitA1 genome is 65,653 bp, which is 23,531 bp larger than the previous prophage WO annotation. Similarly, we identified the new terminal ends of the WOCauB3 prophage (23,099 bp (51%) larger than original estimate of 45,078 bp), extending the previous observation that the end of the genome is beyond the patatin gene[25], along with internal localization of the EAM genes by Sanger sequencing its attP site (domain of unknown function 2426 to recombinase). While we were not able to assemble a complete contig for WOCauB2, it is more than 6,854 bp larger than the original estimate of 43,016 bp, includes multiple ankyrin repeat genes homologous to those in WOVitA1, and, like many other prophage haplotypes (for example, WORiC, WOVitA2 and WOSuziC), integrates directly into Wolbachia's magnesium chelatase (chlI) gene.

**The EAM is enriched with eukaryotic-like domains**. We then analysed each phage WO protein domain for homology and surrounding peptide architecture. Unlike the single-domain architecture of phage WO's structural genes, EAM genes are highly polymorphic and encompass fusions of both eukaryotic-like and bacterial protein domains. By extending the analysis to include homologous prophage regions from all sequenced Wolbachia chromosomes, 10 types of protein domains with putative eukaryotic functions were uncovered spanning four predicted functions: (i) toxins; (ii) host–microbe interactions; (iii) host cell suicide; and (iv) secretion of proteins through the cell membrane (Fig. 2). Notably, over half of these domain types (6/10; latrotoxin C-terminal domain (CTD), PRANC, NACHT, SecA, gwv_1093 N-terminal domain and Octomom-N-terminal domain) share greater amino-acid homology to eukaryotic invertebrates than to bacteria in GenBank. Among this subset with eukaryotic sequence homology, the protein domains are almost exclusively found in the prophage EAM region ($N = 17$) versus the Wolbachia chromosome ($N = 2$). In the latter case, the two chromosomal latrotoxin-CTDs (wNo_10650 and wHa_05390) are flanked by phage-associated genes and transposases, indicating a likely phage WO origin and subsequent genomic rearrangement. This pattern differs from other EAM protein domains with bacterial homology, which are equally dispersed in phage WO ($N = 19$) and the Wolbachia chromosome ($N = 18$) (Fig. 2, Fisher's exact test, $P = 0.0072$). The difference importantly indicates that the eukaryotic-like protein domains are highly enriched in the EAM, suggesting a near exclusive role in phage WO biology.

**The black widow latrotoxin-CTD**. Latrotoxin-CTD is the most prevalent eukaryotic-like domain in prophage WO. Originally described for its major role in the venom of widow spiders (Latrodectus species), latrotoxins act extracellularly to cause the formation of ion-permeable membrane pores in their vertebrate or invertebrate victims. The CTD, specifically, is associated with the latrotoxin precursor molecule (protoxin) and could possibly act intracellularly to facilitate disintegration of the spider's toxin-producing cells[28]. While latrotoxins are generally considered exclusive to spiders, CTD homologues in Wolbachia, Rickettsiella grylli[28], and a transcriptome from a Wolbachia-infected stink bug[29]

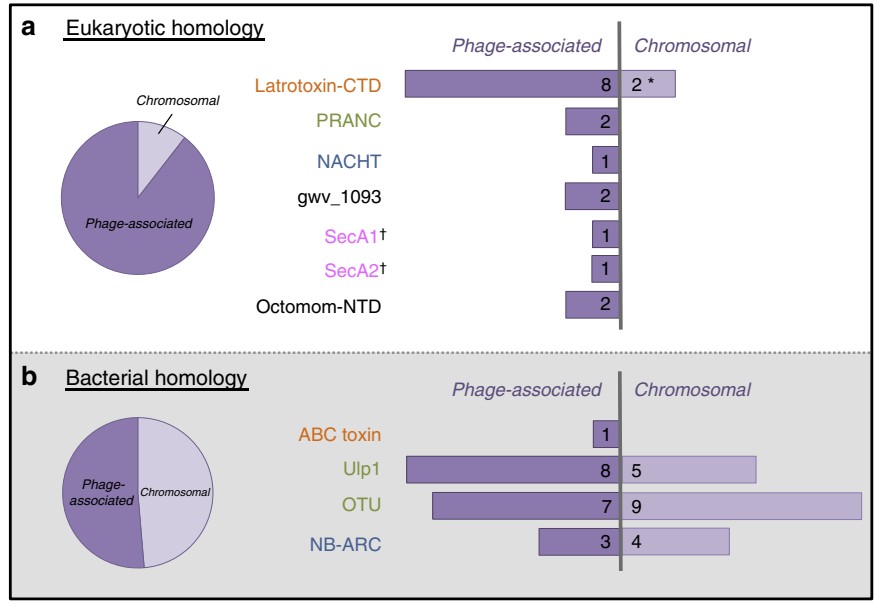

**Figure 2 | Eukaryotic-like EAM genes are enriched in prophage WO regions.** EAM genes with (**a**) eukaryotic homology are most likely to be associated with prophage WO while those with (**b**) bacterial homology are both phage-associated and found scattered throughout the Wolbachia chromosome. *The two chromosomal latrotoxin-CTD domains (wNo_10650 and wHa_05390) are located within phage-associated genes and transposases, indicating a potential genomic rearrangement. †SecA represents one 'domain type' but is listed separately because phage WO contains two different homologues (that is, wHa_3920 and wHa_3930). Putative functional categories are as follows: anti-eukaryotic toxins (orange); host–microbe interactions (green); host cell suicide (blue); secretion of virulence factors (pink); and unknown (black). Octomom refers to WD0513 of the wMel genome.

have been reported. Here phylogenetic analysis implies that the latrotoxin-CTD horizontally transferred between widow spiders and phage WO (Fig. 3). Reciprocal search queries using homologous spider and phage CTDs return the same BLASTP hits shown in Fig. 3. Notably, phage WO CTD sequences have the highest amino-acid similarity to black widow spider homologues that target invertebrates, which are the primary hosts of *Wolbachia*. While convergent evolution could explain amino-acid sequence similarities of these latrotoxin-CTD variants, black widows and *Wolbachia* occur in overlapping ecological niches (*Wolbachia* are known to infect spiders of the family *Theridiidae*) in which gene transfers are likely to happen[30]. We also confirmed the presence of *Wolbachia* in three independent *Latrodectus geometricus* samples by amplifying *Wolbachia* 16S rDNA and *wsp* membrane protein genes. The transfer event was apparently followed by a relatively more recent transfer from phage WO to animals in the *Aedes aegypti* genome, where the region is located between genes of mosquito origin (fibrinogen-related protein (AAEL004156) and GalE3 (AAEL004196)).

**Toxin activation by eukaryotic furin cleavage.** Latrotoxin-CTD is universally located at the 3′-terminal ends of both conserved spider latrotoxin genes[31] and enormous, polymorphic and eukaryotic-like phage WO genes (up to 14,256 bp). There is a high incidence of eukaryotic furin cleavage sites that immediately precede the latrotoxin-CTD. In spiders, cleavage at these sites by the eukaryotic furin protease in the trans-Golgi network or extracellular matrix is required for latrotoxin activation before the toxin exerts its effects on the victim. We show that all prophage WO EAMs contain at least one site for eukaryotic furin cleavage (Supplementary Table 3), and the proportion of all EAM genes with predicted furin cleavage sites (25%) is twofold greater than that of the genes in the core phage genome (11%, Fisher's exact test, $P < 0.0001$), defined as the conserved bacteriophage region from recombinase to patatin. In regards to the phage WO latrotoxin-CTD, its preferential localization in prophage WO genomes versus the rest of the *Wolbachia* chromosome, conservation of eukaryotic furin cleavage sites, large eukaryotic-like length, homology to invertebrate-specific toxins and reduced

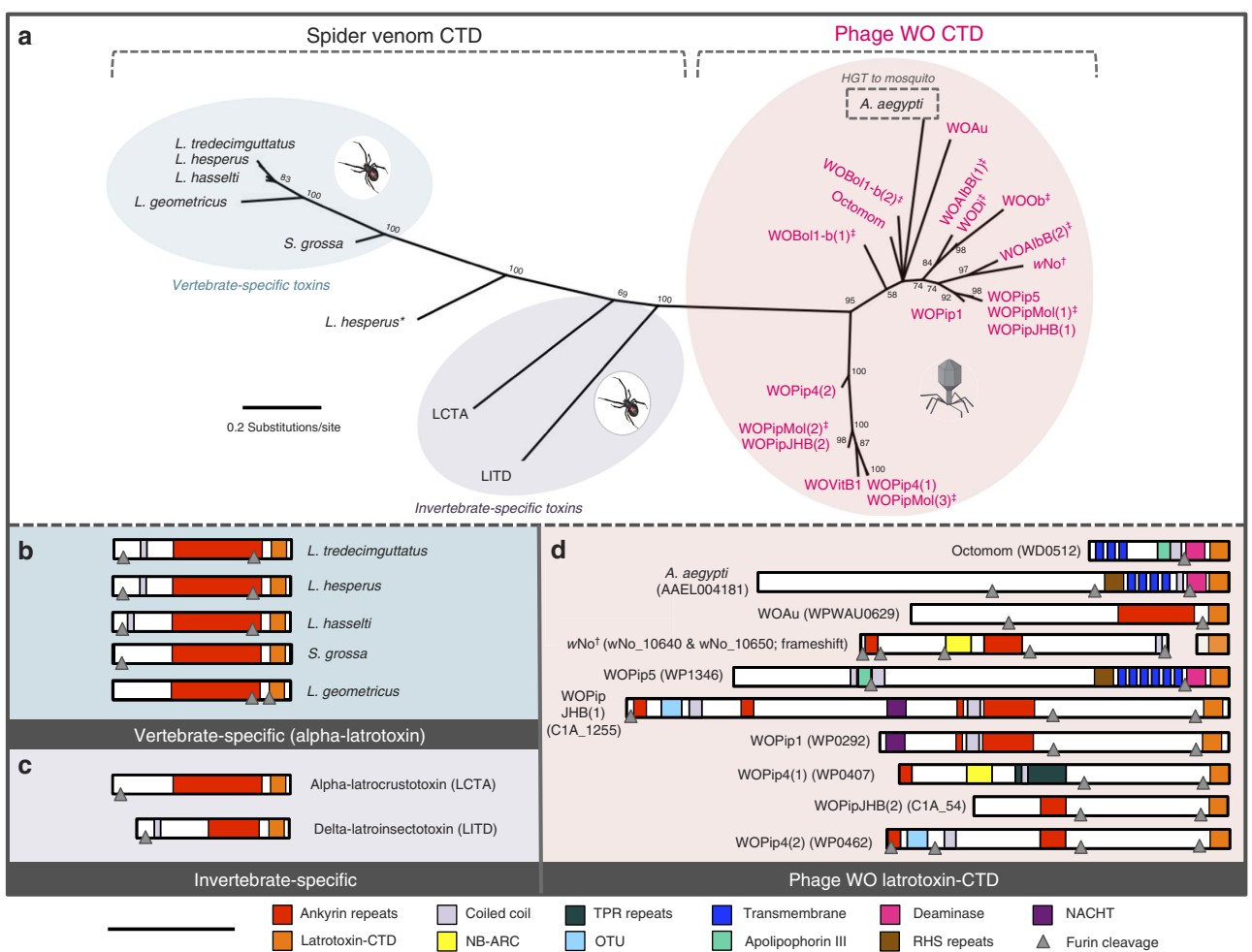

**Figure 3 | Latrotoxin-CTD comparative analyses support lateral genetic transfers. (a)** Phylogeny of phage WO latrotoxin-CTD protein domains and their eukaryotic homologues was constructed by Bayesian analysis of 74 amino acids using the JTT model of evolution. Consensus support values are shown at the nodes. Comparative protein architecture shows that spider venom **(b)** vertebrate-specific alpha-latrotoxins and **(c)** invertebrate-specific alpha- and delta-latrotoxins are highly conserved, whereas **(d)** phage WO are not. WO denotes the specific phage haplotype while genome locus tags are listed in parentheses. Predicted furin cleavage sites, listed in Supplementary Table 3, are illustrated with grey triangles. *A second *L. hesperus* sequence represents a recently described downstream paralogue with unknown toxin activity[32]. †wNo_10650 is located within phage-associated genes and transposases, indicating a potential genomic rearrangement of a phage region. ‡Architecture is not shown for sequences on incomplete contigs (WOBol1-b, WOAlbB, WODi, WOPipMol and WOVitB) because complete peptide information and specific phage association are unknown. Scale bar, 1,000 amino acids.

divergence relative to the spider venom homologues is consistent with a eukaryotic origin and post-translational processing by furin peptidases.

**Pox protein repeats of ankyrin C terminus.** Sequences with homology to predicted domains central in modifying animal proteins are also abundant in the phage WO EAM. For example, homologs of the Pox protein repeats of ankyrin CTD (PRANC) in the WOVitA1 genome (gwv_1092) occur in multiple parasitic wasp hosts (Supplementary Table 4) and their eukaryotic viruses. Reciprocal BLASTP searches retrieve the same best hits and support previous findings that this protein domain horizontally transferred between eukaryotic viruses, animals and *Proteobacteria*[33]. The discovery of a eukaryotic-like PRANC domain in phage WO parallels its presence in the Poxviridae virus family, in which it functions in evasion of eukaryotic immune responses via modification of host ubiquitination. PRANC is related to amino-acid sequences in F-box proteins, which are eukaryotic proteins involved in protein degradation. The PRANC domain also occurs in vaccina virus, ectromelia virus, cowpox virus and Orf virus,

and can regulate nuclear factor-κB signalling pathway to inhibit transcription of inflammatory cytokines[34].

**Conserved ankyrin and tetratricopeptide repeat protein.** Adjacent to the PRANC-encoding gene in WOVitA1's EAM is an ankyrin and tetratricopeptide repeat (TPR)-containing gwv_1093. Ankyrin repeats and TPRs mediate a broad range of protein–protein interactions (apoptosis, cell signalling, inflammatory response and so on) within eukaryotic cells and are commonly associated with effector proteins of certain intracellular pathogens[35,36]. In *Wolbachia*, ankyrins within the core phage genome have been associated with reproductive manipulation of the insect host[37,38]. While generally rare in viral genomes (Supplementary Figs 2 and 3), these repeat regions occur in all prophage WO haplotypes from sequenced *Wolbachia* genomes ($N = 23$). Phylogenetic analysis using reciprocal BLASTP hits (Fig. 4) shows that the N-terminal sequences of the TPR-containing gwv_1093 are embedded within a diverse set of homologues from many arthropod lineages (Fig. 4b), with the most recent transfer putatively occurring between phage WO and *Solenopsis invicta* (Fig. 4c). In this species, the gene is located between ant genes

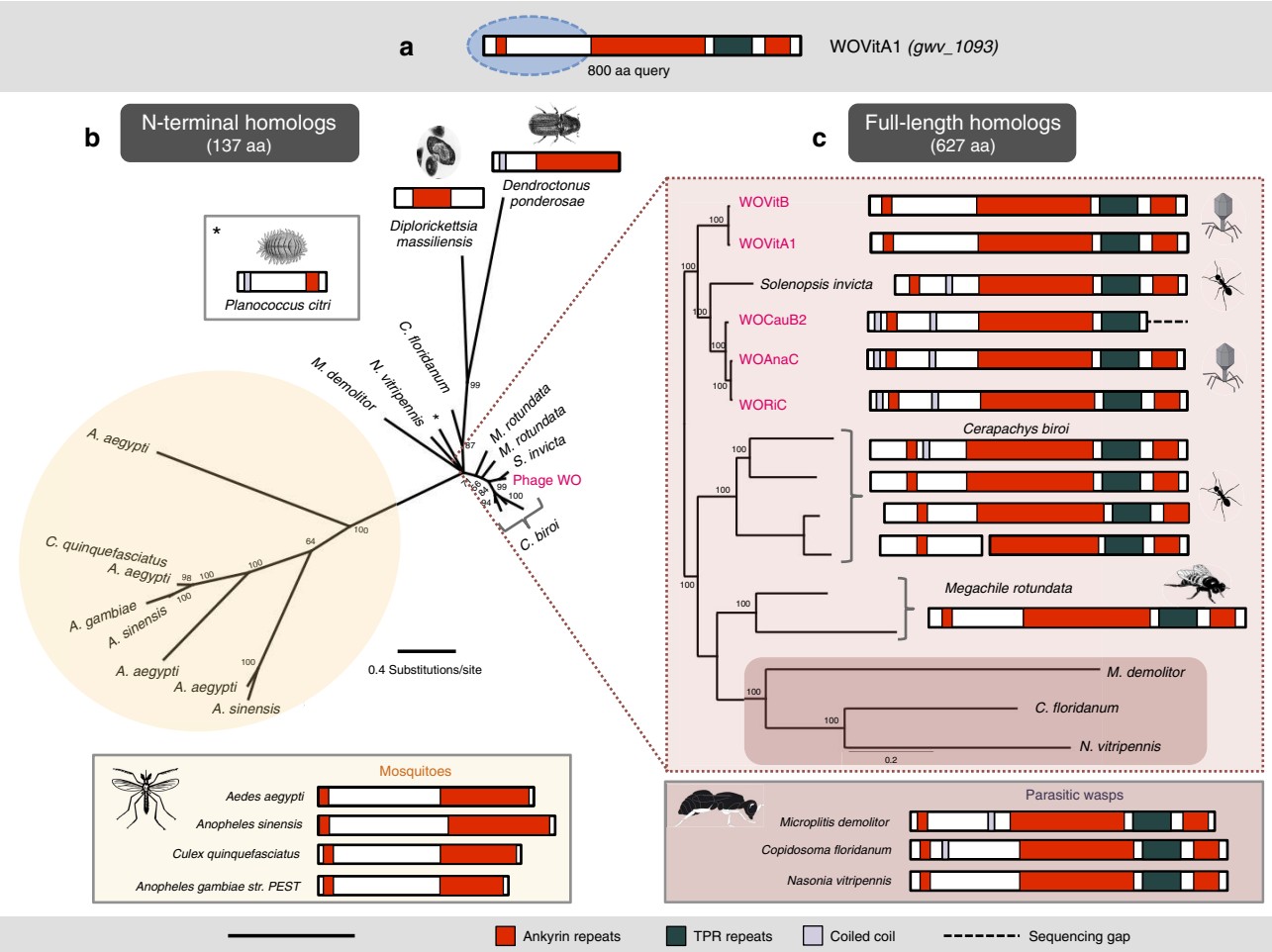

**Figure 4 | Related TPR and ankyrin proteins support lateral genetic transfer.** (**a**) A BLASTP query of WOVitA1's gwv_1093 N terminus reveals homologues in mosquitoes, ants, beetles, a mealy bug, a solitary bee and one obligate intracellular gammaproteobacteria. Bayesian phylogenetic trees were constructed based on (**b**) a 137-aa alignment of all homologues with E-value $< e^{-40}$ using the LG + G model of evolution. (**c**) To resolve taxa closest to phage WO, trees were reconstructed based on a 627-aa alignment of all homologues with an E-value of 0 using the JTT + I + G model of evolution. Isoforms were removed from each alignment. Both trees are unrooted. Consensus support values are shown at the nodes. Chromosomal neighbourhood analyses of available animal genome sequences indicate that animal homologues to the phage WO protein are on contigs with other animal genes. Scale bar, 1,000 amino acids.

bicaudal D and rho guanine nucleotide exchange factor 11. As *S. invicta* can naturally harbour *Wolbachia*[39], either a gene transfer event occurred between these ecologically associated taxa or the *S. invicta* homologue could be an assembly artefact. This ant genome assembly was based on samples from a region rarely infected with *Wolbachia* (Y Wurm, personal communication) and there are no other *Wolbachia*/prophage WO homologues in the *S. invicta* genome; therefore, the latter explanation seems unlikely. Moreover, other gwv_1093 homologues are from insect genome sequences of uninfected strains, that is, *Nasonia vitripennis*, and thus they cannot be derived by an assembly artefact. On the basis of parsimony, the transfer event may have occurred from arthropod to phage WO since the arthropod taxa comprise a more diverse set of lineages. However, the reverse is plausible as transfers from *Wolbachia* to their arthropod hosts are common[40–42].

**NACHT.** Another instance of genetic transfer involves the programmed cell death (PCD) domain, NACHT (Fig. 5). Eukaryotic NACHT-containing proteins are typically engaged in PCD by acting as pathogen sensors and signal transduction molecules of the innate immune system[43]. A polymorphic prophage WO homologue harbours ankyrin repeats and a latrotoxin-CTD directly downstream from a NACHT domain (Fig. 5a). NACHT domains have been identified in animals, fungi and bacteria[44], and phylogenetic patterns indicate multiple instances of horizontal transfer[45]. A NACHT-containing peptide was recently discovered in the *Clostridium difficile*-infecting phage phiCDHM1 (ref. 46). In contrast to prophage WO, it is bacterial in both amino-acid homology and protein architecture. While all BLASTP and reciprocal BLASTP queries of the phiCDHM1 NACHT domain yield only bacterial homologues, BLASTP searches of the prophage WO NACHT domain yield only animal homologues, and reciprocal BLASTP searches of these yield only hits to prophage WO and other animals. Similar to the phylogeny of the N terminus of the TPR-containing gwv_1093, this single NACHT domain sequence in prophage WO is embedded within a more diverse set of homologues in arthropods (Fig. 5b,c). Phylogenetic analyses place the prophage WO variants adjacent to a divergent *Bombyx mori* sequence, though these variants have slightly closer total homology to *Culex quiquefasciatus* mosquitoes that harbour *Wolbachia* with related prophage WO variants.

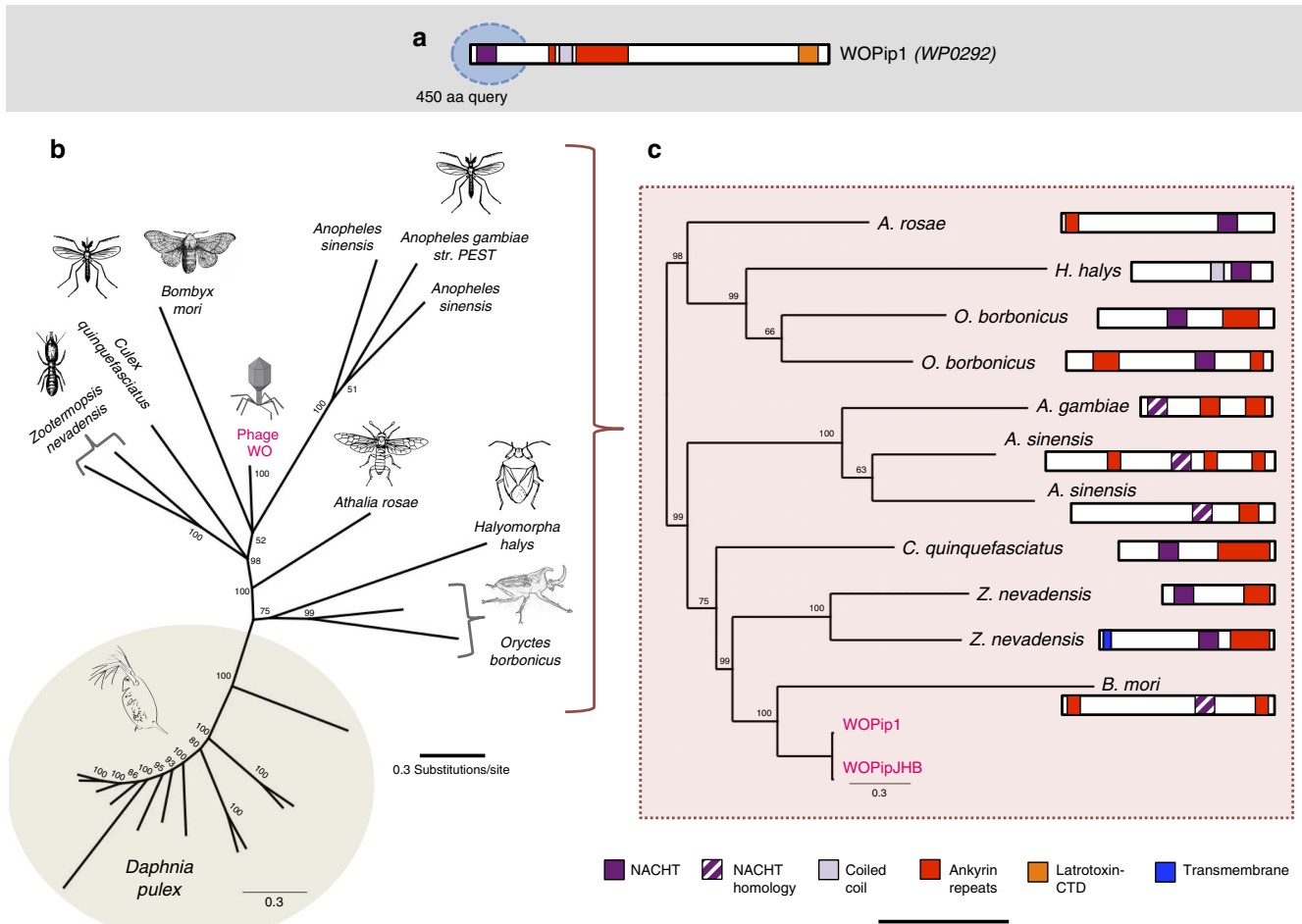

**Figure 5 | Phylogeny and protein architecture of the cell death domain, NACHT.** (**a**) A BLASTP query of prophage WO's NACHT region reveals homologues throughout arthropods and crustaceans. (**b**) Bayesian phylogenetic trees were constructed based on a 271-aa alignment of all homologues with *E*-value $< e^{-15}$ and coverage $> 70\%$ using the cpREV + G model of evolution. To resolve taxa closest to prophage WO, all *Daphnia* sequences were removed from the alignment and clusters of highly divergent residues (that is, five or more sequential residues with <15% pairwise identity) were trimmed. (**c**) Trees were reconstructed based on this 262-aa alignment using the LG + G model of evolution. Consensus support values are shown at the nodes. Both trees are unrooted. Chromosomal neighbourhood analyses of available animal genome sequences indicate that animal homologues to the prophage WO protein are on contigs with other animal genes. Scale bar, 1,000 amino acids.

## Discussion

Metagenomic analysis of the complete genome from phage WO particles reveals all formerly described phage genetic modules (lysogeny, baseplate, head, replication, virulence, tail and patatin-like phospholipase[27]), as well as a new group of genes that we collectively group into a eukaryotic association module (EAM). Some of these genes (i) appear to encode protein domains and cleavage sites central to eukaryotic functions, (ii) occur in both phage and metazoan hosts, (iii) comprise the second largest phage gene to date (14,256 bp) and (iv) are absent from mutualistic, phage-free genomes of *Wolbachia*. Together, these genes increase the phage WO genome size by roughly 50% and include 10 types of protein domains with four predicted eukaryotic functions: toxins; host–microbe interactions; host cell suicide; and secretion of proteins through the cell membrane. Notably, over half of these domain types share greater amino-acid homology to eukaryotic invertebrates than to bacteria in GenBank. Among this subset with eukaryotic sequence homology, the protein domains are almost exclusively found in the phage EAM. An EAM has never before been reported in bacteriophage genomes, to our knowledge, possibly because phages of obligate intracellular bacteria occupy a unique eukaryotic-enclosed niche and are relatively understudied.

The presence of eukaryotic-like protein domains in bacteriophage genomes is of special note as they curiously mirror eukaryotic genes in large eukaryotic viruses that aid in viral mimicry and manipulation of host processes[47,48]. In phage WO, these animal protein domains are central to anti-eukaryotic functions including the black widow latrotoxin, PCD (NACHT), immune evasion (PRANC) and protein–protein interactions.

Bacteriophage WO frequently transfer between *Wolbachia* co-infections in the same animal host[49,50] and to the host genome as part of large transfers of the *Wolbachia* chromosome[40,41]. We previously reported that phage WO in *Wolbachia* of *N. vitripennis* were also capable of transferring adjacent, flanking, non-phage genes in the process of exchange between coinfections[51]. For two of these flanking genes, sequence evidence indicated that *Wolbachia* genomes may be able to receive eukaryotic DNA[42,52,53]. However, the nature of these lateral genetic transfers remained to be elucidated as these regions were not previously known to be part of the packaged phage genome until now. Here we demonstrate that genes with eukaryotic homology are constituents of phage WO and its EAM, and they either retain conservation of eukaryotic furin cleavage sites and a large eukaryotic-like length (that is, latrotoxin-CTD), or they exhibit markedly reduced or no diversity relative to the arthropod homologues as the WO sequences exist as single or a few representatives (NACHT and TPR-containing proteins). Moreover, WO protein domains with eukaryotic homology are highly enriched in the EAM over WO protein domains with bacterial homology. On the basis of this work, we suspect that systematic surveys of phage genomes in intimate host-associated bacteria may uncover a broad range of eukaryotic-like protein domains involved in phage life cycle adaptations and phage–eukaryote interactions. Of particular note is the reported association between phage WO genes, specifically ankyrins, transcriptional regulators and the Ulp1 operon, and *Wolbachia*'s ability to manipulate host reproduction[37,38,54–56].

The mechanisms by which these protein domains are exchanged with phage WO are unknown and could follow at least three models (Fig. 6). First, animal genetic material could directly transfer to and from WO genomes during phage particle propagation in the cytoplasm of animal cells (Fig. 6b) or during packaging inside *Wolbachia* cells that are lysing and exposed to the eukaryotic cytoplasmic environment. Packaging of eukaryotic host RNAs, for instance, occur in the virions of herpesvirus[57] and cytomegalovirus[58]. Second, genes may transfer between animal genomes and the *Wolbachia* chromosome and then to prophage WO. For this scenario to be plausible, animal genetic material transferred in random locations in the *Wolbachia* genome would have to be preferentially lost in non-phage-associated locations from the *Wolbachia* chromosome (Fig. 6c) because domains with eukaryotic homology are highly enriched in the phage/prophage WO EAM versus the rest of the chromosome (Fig. 2). Third, DNA may transfer first between animal genomes and intermediary entities, such as eukaryotic viruses or other obligate intracellular bacteria, and then to phage WO and/or *Wolbachia* (Fig. 6d). In fact, the PRANC-domain (described in gwv_1092) was named for its discovery in and association with eukaryotic Pox viruses. Finally, once DNA is incorporated into a prophage genome, it is susceptible to recombination with other phage WO haplotypes located in the same *Wolbachia* chromosome and can transfer from one haplotype to another.

Alternatively, these protein domains could originate in the phage or Wolbachia prophage and be particularly prone to transfer, maintenance and spread in their recipient arthropod genomes (Fig. 6b,c). For this scenario to be plausible, it would have to imply that phage WO genetic material independently and repeatedly transfers to arthropods and spreads through the host population, which would subsequently be followed by loss of these phage genes or recombination with other non-transferred phage genetic material so that the eukaryotic sequence variation clusters separately from the phage WO sequence(s). Each mode of transfer is possible, though the eukaryotic length of these genes, presence of furin protease domains and enrichment in the phage WO EAM provide tentative evidence for their eukaryotic origin.

Why are these protein domains present in the EAM of bacteriophage WO? Some phages of obligate intracellular bacteria may have to overcome two major challenges not encountered by the well-studied phages of free-living bacteria. First, they are contained within both bacterial and eukaryotic membranes, posing an enigmatic 'twofold cell challenge'. They may not only have to breach peptidoglycan and permeabilize bacterial membranes but they may also have to exit (and enter) across the eukaryotic membrane(s) that directly encapsulates the bacteria. Second, like their bacterial hosts, they must survive the internal cellular environment of the animal host, including the innate immune response and autophagy, while searching for phage-susceptible bacteria. Phage WO can dwell in the eukaryotic cytoplasm and extracellular matrix that they encounter on bacterial lysis[26], raising the likelihood of direct interaction with host membranes and intracellular biology. In this context, EAM protein domains are prime candidates to aid in functions, including cell lysis (latrotoxin-CTD), manipulation of PCD (NACHT and NB-ARC), host ubiquitination (OTU and Ulp1), insecticidal toxicity (ABC toxin) and interaction with host proteins (ankryin repeats and TPRs). Rather than simply act as virulence factors to benefit their bacterial host, their massive proportion of genomic real estate (up to 60% of the prophage genome, Supplementary Fig. 4) implies that they may be necessary to phage biology and likely have a direct impact on phage propagation. The concept of phage-mediated ecosystem modification as an alternative to bacterial virulence is not new[59] but, much like the biology of phage WO, is relatively understudied.

Phage WO is not the only virus described within obligate intracellular bacteria. Bacteriophage also infect *Chlamydia*, yet still do not directly contend with the eukaryotic membrane. Rather, they attach to dormant chlamydial cells (that is, reticulate bodies) and enter via phagocytosis or endocytosis of the bacteria[60]. The phages then alter development of their bacterial host, which leads to disintegration of the chlamydial

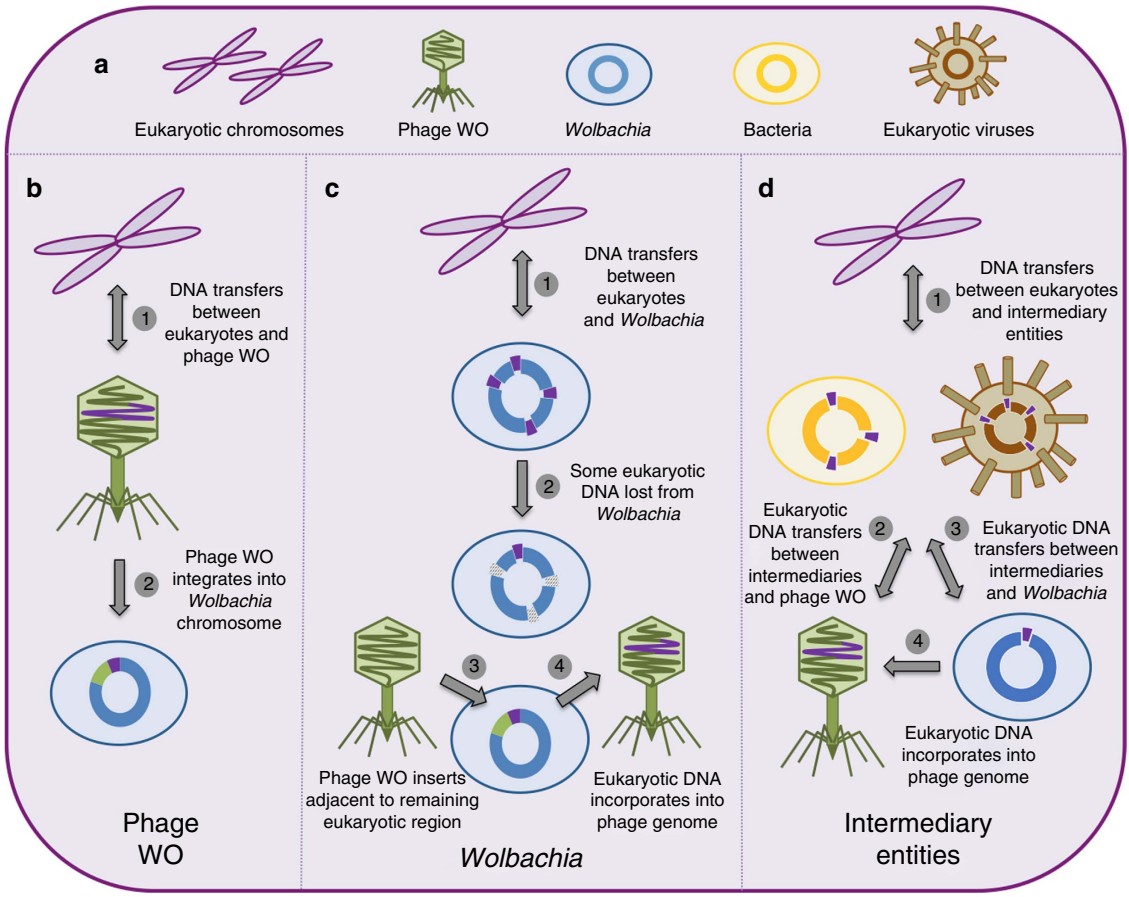

**Figure 6 | Models of lateral DNA transfer between eukaryotes and bacteriophages.** (**a**) The eukaryotic cell can harbour multiple microbes capable of horizontal gene transfer. Genetic transfers between eukaryotes and bacteriophages can, in theory, occur (**b**) directly between eukaryotic chromosomes and phage genomes; (**c**) indirectly between eukaryotic and *Wolbachia* chromosomes; or (**d**) indirectly between eukaryotic chromosomes and intermediary entities, such as eukaryotic viruses and other intracellular bacteria.

inclusion and subsequent lysis of the eukaryotic host cell[61,62]. The nature of phage WO's lifestyle, on the other hand, may require a distinct interaction with multiple membranes and immune responses because lytic activity of phage WO has been associated with typical bacterial cell defects, including degraded bacterial DNA, a detached inner membrane and exit of the phage particles from inside *Wolbachia* and its host cell into the extracellular matrix of the reproductive tissues[26]. Bacteriophages of free-living bacteria also regularly colonize eukaryotic environments, particularly those associated with mucosal surfaces[63]. They, however, do not infect or traverse the eukaryotic membrane and are still within the genomic boundaries of the bacterial virosphere.

Temperate double-stranded DNA phages also occur in facultative symbionts of aphids[64] and tsetse flies[65]. While *Wolbachia* has never successfully been cultured outside of host cells[66], these facultative symbionts can replicate both intra- and extracellularly (JW Brandt, personal communication) suggesting that their phages are not constrained by the same twofold cell challenge. In addition, their phages encode a traditional lytic cassette (holin and lysozyme) that correlates with the need to deal with bacterial membranes. In some cases, the phages harbour bacterial-derived toxins that target eukaryotic cells[67], and these function mutualistically in aphids by arresting development of parasitoid wasp larvae[64]. Furthermore, unlike phage WO that is stably maintained in the lab, these phages are readily lost in the absence of parasitoids during laboratory rearing, presumably due to the cost of their toxins[68].

In addition to providing new insights into the evolution of bacteriophages and showing phage WO genomes to be far more complex than previously described, the findings here reveal the bacteriophage WO attachment and bacterial integration sites as well as evidence for gene sharing between metazoan hosts and phages of obligate intracellular bacteria. We suggest that the possible acquisition and retooling of intact eukaryotic-like domains in phage WO could be viewed as analogous to the commandeering of host genes by eukaryotic viruses. Whether lateral genetic transfers between metazoans and bacteriophages are common in the symbiotic virosphere remains to be determined.

## Methods

**Insect and bacterial strains.** The transfected line of the Mediterranean flour moth *Ephestia kuehniella* harbouring *Wolbachia* strain *w*CauB was obtained with the help of Takema Fukatsu and Tetsuhiko Sasaki[22]. Moths were maintained at 24 °C and 70% humidity on a diet consisting of wheat bran, glycerol and dried yeast (20:2:1 w/w). The introgressed line of the parasitoid wasp *N. giraulti* harbouring *Wolbachia* strain *w*VitA, termed IntG12.1, was previously derived by repeatedly backcrossing *N. vitripennis* (strain 12.1) females to *N. giraulti* males for nine generations[69]. The strain was incubated at 25 °C using the flesh fly *Sarcophaga bullata* as host.

**Phage particle purification.** Phage particles were isolated according to Fujii *et al.*[22] with modifications. Approximately 4 g of adult insects were homogenized in 29.6 ml cold SM buffer (50 mM Tris-HCl, pH 7.5, 0.1 M NaCl, 10 mM MgSO$_4$.7H$_2$0 and 0.1% (w/v) gelatin). NaCl and RNase A were added to a final concentration of 1 M and 1 µg ml$^{-1}$, respectively. The homogenate was incubated on a shaker at 4 °C for 1 h and then centrifuged at 13,000$g$ for 10 min at 4 °C. Polyethylene glycol 6000 was added to a final concentration of 10% to precipitate phage particles, incubated at 4 °C for 1 h with gentle shaking and centrifuged at 13,000$g$ for 10 min.

The pellet was resuspended in 5 ml TM buffer (50 mM Tris-HCl, pH 7.5 and 10 mM MgCl$_2$.6H$_2$O) and mixed with an equal volume chloroform. The suspension was centrifuged at 3,000$g$ to remove polyethylene glycol, and the aqueous phase was filtered through a 0.22 μm filter to remove bacterial cells. The suspension was centrifuged at 60,000$g$ for 1 h at 4 °C to collect phage particles. The pellet was suspended in 10 μl TM buffer.

**Phage DNA extraction and metagenomic sequencing.** The phage suspension was treated with RQ1 RNase-Free DNase (Promega) for 30 min at 37 °C, followed by heat inactivation for 10 min at 65 °C, to remove host contamination. Phage DNA was extracted from the suspension using the QIAamp MinElute Virus Spin Kit (Qiagen) and amplified using the REPLI-g Mini Kit (Qiagen). Following amplification, paired-end DNA libraries were prepared according to the manufacturer's (Illumina) instructions, and samples were sequenced with an Illumina HiSeq 2000 (2 × 100-nucleotides read length).

**Bioinformatics and statistics.** Metagenomic sequences (reads) were trimmed, paired and assembled into contigs using the CLC Assembler (CLC bio) with bubble size = 50, insertion and deletion cost = 3, mismatch cost = 2, length fraction = 0.6, minimum contig size = 130, similarity = 0.5, minimum distance = 90 and maximum distance = 200. Contigs were compared with the GenBank non-redundant database using NCBI's BLASTN (http://blast.ncbi.nlm.nih.gov/Blast.cgi) and those with similarity to phage WO and/or *Wolbachia* ($E$-value $< 10^{-10}$) were manually annotated using Geneious (Biomatters Ltd.). Individual reads were mapped to reference sequences using Geneious. Open reading frame homology searches were performed to determine putative function using NCBI's BLASTP (http://blast.ncbi.nlm.nih.gov/Blast.cgi) and Wellcome Trust Sanger Institute's pfam database (http://pfam.sanger.ac.uk). Coiled coil domains were predicted with EMBL's Simple Modular Architecture Research Tool (SMART, http://smart.embl-heidelberg.de). Furin cleavage sites were identified using PiTou (http://www.nuolan.net/reference.html). The number of genes with and without furin cleavage sites was analyzed with respect to phage-region using Fisher's exact test (GraphPad Software). Phylogenetic trees were built using the Bayes plugin in Geneious and model selection for each Bayes analysis was estimated using ProtTest[70].

**Confirmation of phage WO terminal genes.** Genomic DNA was extracted from *w*VitA-infected *N. vitripennis* (strain 12.1) and *w*CauB-infected *E. kuehniella* individuals using the Gentra Puregene Tissue Kit (Qiagen). Primers were designed for both WOVitA1 and WOCauB3 *attP* sites, respectively: VitA1_attF (5′-CGAA GAACCAGCACAGGGTGG-3′); VitA1_attR (5′-GCTGGAAGAGGGCATCTG CATC-3′); CauB3_attF (5′-TCGTGACTGCCCTATTGCTGCT-3′); and CauB3_attR (5′-ATGCGGCCAAAGCTGGGTGT-3′). Amplification was performed in a Veriti thermal cycler (Applied Biosystems) using GoTaq green master mix (Promega) under the following conditions: 94 °C for 2 min; 35 cycles of 94 °C for 30 s, 53 °C for 30 s, 72 °C for 1 min; and a final elongation cycle of 72 °C for 10 min. PCR products were sequenced via Sanger sequencing (Genewiz, Inc).

**Data availability.** The phage WOVitA1 genome assembly reported in this paper has been deposited in NCBI under accession number KX522565. The *N. vitripennis* viral metagenome sequences have been deposited in the SRA under accession number SRR3560636 and BioProject PRJNA321548. The *w*CauB-infected *E. kuehniella* viral metagenome sequences have been deposited in the SRA under accession number SRR3536639 and BioProject PRJNA321549. Data referenced in this study are available in NCBI with accession codes AE017196 (*w*Mel), AM999887 (*w*Pip), CTEH00000000 (*w*PipMol), ABZA00000000 (*w*PipJHB), CP001391 (*w*Ri), CAOU00000000 (*w*Suzi), AMZJ00000000 (*w*Di), AAGB01000001 (*w*Ana), CAGB00000000 (*w*AlbB), CAOH00000000 (*w*Bol1-b), JYPC00000000 (*w*Ob), CP003884 (*w*Ha), CP003883 (*w*No), LK055284 (*w*Au), AP013028 (*w*Cle), HE660029 (*w*Oo), PRJNA213627 (*w*VitA), AB478515 (WOCauB2), AB478516 (WOCauB3), KC955252 (WOSol), HQ906665 and HQ906666 (WOVitB).

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

## Acknowledgements

We thank Tetsuhiko Sasaki and Takema Fukatsu for assistance with providing the *w*CauB-infected moths; Rini Pauly for computational assistance; Merri Lynn Casem for providing widow spiders; Kerry Oliver for information on *Hamiltonella*; Yannick Wurm for information on *Solenopsis*; and Michael Gerth, Julie Dunning Hotopp, Kevin Kohl and Jason Metcalf for feedback on the manuscript. We apologize to our colleagues for not being able to include all possible references due to citation restrictions. This research was funded by NIH Awards R01 GM085163 and R21 HD086833 and NSF Awards DEB 1046149 and IOS 1456778 to S.R.B. The funders had no role in study design, data collection and interpretation or the decision to submit the work for publication.

## Author contributions

Sarah R. Bordenstein designed and performed the experiments, analysed the data, prepared figures and tables, and wrote and reviewed drafts of the paper; Seth R. Bordenstein conceived and helped design the experiments, analysed the data, and wrote and reviewed drafts of the paper.

## Additional information

**Competing financial interests:** The authors declare no competing financial interests.

**How to cite this article**: Bordenstein, S. R. & Bordenstein, S. R. Eukaryotic association module in phage WO genomes from *Wolbachia*. *Nat. Commun.* **7,** 13155 doi: 10.1038/ncomms13155 (2016).

