## [Peer Review File · Nature Communications]

Reviewer #1 (Remarks to the Author)

This manuscript reports eukaryotic protein domains in bacteriophages (WO) of Wolbachia, a group of alpha-proteobacteria that commonly infect arthropods. These domains often contain important eukaryotic functions and they are encoded in a specific region of the WO genomes (so called Eukaryotic Association Module or EAM). The authors conclude that these domains of eukaryotic origin may allow WO phages to traverse both bacterial and eukaryotic membranes ("two-fold cell challenge").

In general, I think the manuscript is well written and results are interesting, but the discussion lacks sufficient depth. I have the following comments for the authors to consider in their revision.

1. The authors speculate that the acquisition of these eukaryotic domains might have resulted from the "two-fold cell challenge". This is a main conclusion of this study, but the authors basically just wrote one sentence to explain. It will help if the author could further elaborate.
2. Assuming all WO phages need to deal with the membranes of both bacteria and eukaryotes, if the eukaryotic domains are indeed related to this challenge, why aren't they found in all these phages?
3. From the manuscript, it appears that multiple WO phages may infect the same Wolbachia cell. If so, it is important to discuss the possibility of gene transfer between phages within the bacterial cell. If the possibility of transfer between phages cannot be excluded, it will likely change the main conclusion of the finding, even though EAM might ultimately be derived from eukaryotes.

Reviewer #2 (Remarks to the Author)

In this manuscript, "Lateral Genetic Transfers Between Eukaryotes and Bacteriophages" by Bordenstein and Bordenstein (NCOMMS-16-05723), the authors provide evidence for a eukaryotic association module (EAM) in the genomes of bacteriophages. The bacteriophages studied infect bacterial symbionts (i.e., Wolbachia) of insects. It is suggested that the EAMs, encoding proteins with domains that have been predominately associated with eukaryotes, such as NACHT, ankyrin repeats, and latrotoxin-CTD, arose through lateral gene transfer and for which the authors suggest two possible models.

The findings are direct, conclusive and interesting. The manuscript should be of interest to specialists of bacteriophages, Wolbachia, symbiosis, and also of interest to a broader audience in evolutionary biology, genome evolution, and gene transfer events. The writing is generally clear and the data and figures are well documented and formatted.

The experimental approach is one primarily of phage isolation and DNA sequencing, PCR and sequencing of attachment sites, identification and sequencing of bacterial attB sites, and bioinformatics analysis of the sequenced genomes, including BLASTP and phylogenetic tree analysis of related proteins. New insights on the prevalence and phylogeny of EAM gene clusters in Wolbachia phages are obtained, leading to the main conclusion, as stated in the title, of lateral gene transfer from insect to phage.

Some specific comments and suggestions:

1. The Title - "Lateral genetic transfers...", why not the simpler, commonly used "Lateral gene transfer....."?
2. To the non-specialist, the nomenclature for phage and Wolbachia strains could be clarified at the start in the text, in an abbreviation footnote, or in a table (supplemental or included). That is, something like "wVitA refers to the Wolbachia strain and WOVitA1 refers to the phage infecting wVitA"....
3. In the Summary, the long sentence of lines 17 - 22 could be two, or needs more clarity as it

stands. Line 18 discover > discovered; proteins domains > protein domains; line 19 delete intact; line 22... various protein domain families to just >various protein domains. Line 27 delete canonical. For my preference, there are more adjectives used in the manuscript than needed to make the clear points.

4. Line 70 - are resolved > were resolved.

5. Line 77 - define patatin (not commonly understood)--- i.e., patatin phospholipase.

6. Line 120 - ...the EAM region.. > the phage EAM region...

7. Line 135, throughout - confirm at the NCBI site- BLASTp > BLASTP

8. Line 229 - They have never before been reported.....because phages have naturally been overlooked..... I'd consider re-wording this sentence, regarding what has been "naturally overlooked". Domains of predominately eukaryotic proteins have been observed in phage proteins.

9. An uncited 2012 paper (Pinchon et al. (PMID:22497736)) discusses Ankyrin repeats in WO phage genomes. This earlier observation appears to be relevant for citation either in the Results paragraph starting at line 182, or the Discussion around line 227. This is also relevant to comment #8, although the authors may suggest a different domain lineage that supports their line 229 statement.

The Methods, Results, Discussion and Supplemental materials yielded no additional comments or suggestions.

Reviewer #3 (Remarks to the Author)

In this remarkable paper by Bordenstein and Bordenstein they describe the discovery of a new class of phage genes that have been captured from eukaryotes. The phage infect bacteria, which themselves infect eukaryotes. The phage have acquired genes from the eukaryotic host that the phage host parasitizes.

Bordenstein and Bordenstein identified the ends of the phage by sequencing across the attP regions from phages, and used this new information to accurately annotate the locations of the prophages in the Wolbachia genome. In doing so, they identified a very large gene (upto 14kb) that contain domains that are predicted to be involved in phage exit from both the prokaryotic and eukaryotic cells that surround the host. The latrotoxin domain that has been identified on the phage genome, for example, is used to open pores in cellular membranes. The authors have identified cleavage sites for eukaryotic proteases and proteins that aid in immune response resistance.

Bordenstein and Bordenstein propose two models for the transfer of DNA into the phage. Their first (direct) model assumes that the phages are being packaged either in the eukaryotic cell's cytoplasm (which seems unlikely since the packaging mechanism relies on prokaryotic proteins) or in partially lysed cells. This latter is also unlikely as usually phages are packaged before the prokaryote is lysed. Is it possible that Wolbachia takes up DNA from its host (e.g as a nutrient) and then packaged with the host? Is it possible that the phage bind to extracellular host DNA or bacteria take up extracellular host DNA (which is often secreted as an immune response, eg. from neutrophils).

This is a remarkable discovery because no one has shown how phages that infect intracellular pathogens can find new hosts. There are many examples of phages infecting intracellular pathogens, including Chlamydia and Listeria. It is highly likely that others will follow Bordenstein and Bordenstein's work and identify similar proteins on those phages (as they note at lines 240-243).

This paper has opened a new research area in biology, the results are revolutionary, and I recommend that it be published in Nature

Minor comments and questions

- Does the latrotoxin work from the inside of a cell outwards as well as from the outside of a cell inwards. The phages are doing the former, but the spiders are doing the latter.
- How does the phage get back into a cell (presumably by attachment on an extracellular Wolbachia)
- Is there evidence of other eukaryotic genes on phages (notably internalins or actA from *Listeria monocytogenes*)
- line 250 transfer is a spelling error
- line 332 μ l should use a greek letter (μ), and here and elsewhere (e.g. lines 370-) there are a couple of instances of the degree symbol missing preceding a C

Reviewer #1

In general, I think the manuscript is well written and results are interesting, but the discussion lacks sufficient depth. I have the following comments for the authors to consider in their revision.

1. The authors speculate that the acquisition of these eukaryotic domains might have resulted from the "two-fold cell challenge". This is a main conclusion of this study, but the authors basically just wrote one sentence to explain. It will help if the author could further elaborate.

We thank the reviewer for the thoughtful feedback and have now revised the manuscript to provide an extensive discussion. The following text has been added on page 22:

"Why are these protein domains present in the EAM of bacteriophage WO? Some phages of obligate intracellular bacteria may have to overcome two major challenges not encountered by the well-studied phages of free-living bacteria. First, they are contained within both bacterial and eukaryotic membranes, posing an enigmatic "two-fold cell challenge". They may not only have to breach peptidoglycan and permeabilize bacterial membranes, but they may also have to exit (and enter) across the eukaryotic membrane(s) that directly encapsulates the bacteria. Second, like their bacterial hosts, they must survive the internal cellular environment of the animal host, including the innate immune response and autophagy, while searching for phage-susceptible bacteria. Phage WO can dwell in the eukaryotic cytoplasm and extracellular matrix that they encounter upon bacterial lysis²⁶, raising the likelihood of direct interaction with host membranes and intracellular biology. In this context, EAM protein domains are prime candidates to aid in functions including cell lysis (latrotoxin-CTD), manipulation of programmed cell death (NACHT and NB-ARC), host ubiquitination (OTU and Ulp1), insecticidal toxicity (ABC toxin) and interaction with host proteins (ankryin repeats and TPRs). Rather than simply act as virulence factors to benefit their bacterial host, their massive proportion of genomic real estate (up to 60% of the prophage genome, Supplementary Fig. 4) implies that they may be necessary to phage biology and likely have a direct impact on phage propagation. The concept of phage-mediated ecosystem modification as an alternative to bacterial virulence is not new⁵⁸ but, much like the biology of phage WO, is relatively understudied."

2. Assuming all WO phages need to deal with the membranes of both bacteria and eukaryotes, if the eukaryotic domains are indeed related to this challenge, why aren't they found in all these phages?

The EAM is indeed present in every complete phage WO haplotype, and we added Supplementary Information Fig. 4 (in addition to the existing Supplementary Information Table 2) to clarify/illustrate their presence. However, not all individual eukaryotic domains are universally conserved across the haplotypes. While a transcriptional regulation cluster and one or more ankyrin repeat containing proteins are present in all EAMs, other protein domains and genes are inconstant. For example, the wPip *Wolbachia* genome contains five prophage regions with EAM genes that are genetically different. Variability in the EAM region is consistent with the independent evolution of each phage WO haplotype and high rates of gene loss and gain in phage WO (Kent et al. 2011, PLOS One). It could be possible that each phage overcomes the cellular challenges with its own unique recipe of effectors or that they synchronize lysis and hijack proteins from other phages.

3. From the manuscript, it appears that multiple WO phages may infect the same *Wolbachia* cell. If so, it is important to discuss the possibility of gene transfer between phages within the bacterial cell. If the possibility of transfer between phages cannot be excluded, it will likely change the main conclusion of the

finding, even though EAM might ultimately be derived from eukaryotes.

Yes, the reviewer is correct that multiple phages can infect the same *Wolbachia* cell. Based on analyses of conserved structural genes throughout prophage regions, homologous recombination and gene transfer between phage WO haplotypes can occur. However, as noted above, the EAM genes tend to be more variable than the core genome, even among prophage WO haplotypes in the same *Wolbachia* genome. While we do not believe this changes the conclusion of lateral genetic transfers between eukaryotes and the phage genome, we do agree that phage-to-phage transfer is relevant when discussing all the possible gene transfer events. The following statement has been added on page 19:

"Finally, once DNA is incorporated into a prophage genome, it is susceptible to recombination with other phage WO haplotypes located in the same *Wolbachia* chromosome and can transfer from one haplotype to another".

Reviewer #2

The findings are direct, conclusive and interesting. The manuscript should be of interest to specialists of bacteriophages, Wolbachia, symbiosis, and also of interest to a broader audience in evolutionary biology, genome evolution, and gene transfer events. The writing is generally clear and the data and figures are well documented and formatted.

We appreciate the positive feedback.

Some specific comments and suggestions:

1. The Title - "Lateral genetic transfers...", why not the simpler, commonly used "Lateral gene transfer....."?

Given the cumulative feedback from reviewers, we went further in changing the title to "Novel Eukaryotic Association Module in Phage WO Genomes from *Wolbachia*" to emphasize the discovery of the EAM in the first genome sequences from phage WO particles. We suspect that "phage WO genome" and "eukaryotic association module" will become future search terms.

2. To the non-specialist, the nomenclature for phage and *Wolbachia* strains could be clarified at the start in the text, in an abbreviation footnote, or in a table (supplemental or included). That is, something like "wVitA refers to the *Wolbachia* strain and WOVitA1 refers to the phage infecting wVitA"....

We wholeheartedly agree and have included a supplemental table to alleviate any confusion relating to abbreviations for *Wolbachia* strains and phage haplotypes.

3. In the Summary, the long sentence of lines 17 - 22 could be two, or needs more clarity as it stands. Line 18 discover >discovered; proteins domains > protein domains; line 19 delete intact; line 22... various protein domain families to just >various protein domains. Line 27 delete canonical. For my preference, there are more adjectives used in the manuscript than needed to make the clear points.

Thank you for the suggestions. The sentence from line 17 has been broken into two and all additional changes were incorporated into the summary.

4. Line 70 - are resolved > were resolved.
5. Line 77 - define patatin (not commonly understood)--- i.e., patatin phospholipase.
6. Line 120 - ...the EAM region.. > the phage EAM region...
7. Line 135, throughout - confirm at the NCBI site- BLASTp > BLASTP

We appreciate the edits. All recommendations have been incorporated.

8. Line 229 - They have never before been reported.....because phages have naturally been overlooked..... I'd consider re-wording this sentence, regarding what has been "naturally overlooked". Domains of predominately eukaryotic proteins have been observed in phage proteins.

Thank you for this consideration. We revised the sentence:

"An EAM has never before been reported in bacteriophage genomes, possibly because phages of obligate intracellular bacteria occupy a unique eukaryotic-enclosed niche and are relatively understudied."

9. An uncited 2012 paper (Pinchon et al. (PMID:22497736)) discusses Ankyrin repeats in WO phage genomes. This earlier observation appears to be relevant for citation either in the Results paragraph starting at line 182, or the Discussion around line 227. This is also relevant to comment #8, although the authors may suggest a different domain lineage that supports their line 229 statement.

Thank you for recommending this study. We cited the paper and included the following statement on page 13:

"In *Wolbachia*, ankyrins within the core phage genome have been associated with reproductive manipulation of the insect host^{36,37}."

The statement on line 229, which has been modified, specifically referenced domains with evidence of lateral gene transfer between phage WO and eukaryotes. While the ankyrin motif may possibly be eukaryotic, it is also fairly abundant in bacteria. We found no evidence of direct transfer between phages and insect hosts for *pk1* and *pk2*. However, we do realize the importance of including these studies, as well those looking at the transcriptional regulators and the Ulp1 operon, especially with regard to phage WO and its interaction with the eukaryotic host. These references have been incorporated into the discussion on page 19.

The Methods, Results, Discussion and Supplemental materials yielded no additional comments or suggestions.

Reviewer #3

This is a remarkable discovery because no one has shown how phages that infect intracellular pathogens can find new hosts. There are many examples of phages infecting intracellular pathogens, including Chlamydia and Listeria. It is highly likely that others will follow Bordenstein and Bordenstein's work and identify similar proteins on those phages (as they note at lines 240-243).

This paper has opened a new research area in biology, the results are revolutionary, and I recommend that it be published in Nature

We appreciate the positive feedback. Because phages of intracellular bacteria are particularly challenging to study, we hope that this work sheds light on other systems and encourages special consideration of “prophage boundaries” in genomic studies. We suspect the role of phages in a tripartite system has been greatly underestimated and look forward to future studies.

Minor comments and questions

- Does the latrotoxin work from the inside of a cell outwards as well as from the outside of a cell inwards. The phages are doing the former, but the spiders are doing the latter.

This is a thoughtful question and one that was not fully discussed in the text.

To date, the function of the latrotoxin-CTD has not been resolved. However, before the latrotoxin acts on its victim, it must first be produced in the spider’s cells and released via holocrine secretion. Zhang et al., 2012 suggested that the CTD might aid in the induction of cellular disintegration by associating with the cell membrane (from the inside) via its hydrophobic helix. Therefore, it is likely that the phage CTD functions similar to the spider CTD (both internally). Once the protoxin is released from the spider’s producing cells, the CTD is cleaved via furin protease to form the active toxin, where it acts extracellularly.

We revised the text on page 9 to include a more thorough explanation:

“Originally described for its major role in the venom of widow spiders (*Latrodectus* species), latrotoxins act extracellularly to cause the formation of ion-permeable membrane pores in their vertebrate or invertebrate victims. The CTD, specifically, is only associated with the latrotoxin precursor molecule (protoxin) and could possibly act intracellularly to facilitate disintegration of the spider’s toxin-producing cells.²⁸”

- How does the phage get back into a cell (presumably by attachment on an extracellular Wolbachia)

Unfortunately, we do not know the mode of cellular entry. Prior to this work, the mode of eukaryotic cellular entry by a bacteriophage was unexplored. Now that we have identified the EAM and broadened the scope of potential candidate proteins, we can begin to study and understand the specific phage WO lifestyle.

- Is there evidence of other eukaryotic genes on phages (notably internalins or actA from *Listeria monocytogenes*)

To our knowledge, no other study has identified genes of eukaryotic origin in phages. We agree that a study of the proteins on phage WO’s capsid would be most interesting and could illuminate potential mode(s) of cellular entry.

- line 250 transfer is a spelling error

- line 332 ul should use a greek letter (μ), and here and elsewhere (e.g. lines 370-) there are a couple of instances of the degree symbol missing preceding a C

Thank you; these changes have been incorporated.

REVIEWERS' COMMENTS:

Reviewer #1 (Remarks to the Author):

The authors have addressed my comments in this revision. I don't have additional comments.

Reviewer #2 (Remarks to the Author):

The Authors have thoroughly addressed all previous comments and suggestions provided by this reviewer. Clarification of scientific content throughout and modest expansion of the Discussion provide for an improved manuscript. I have no additional review comments.

Reviewer #3 (Remarks to the Author):

I stand by my previous assessment that this is an exceptional paper that will appeal to a wide audience and has really set a new paradigm for phage interactions with obligate intracellular bacteria. I believe that this work will have a major impact in the work on several important pathogens (as well as Wolbachia!).

I appreciate the authors thoughtful responses to the questions and comments from the reviews, and have no additional concerns about the paper.